# Rapid global ocean-atmosphere response to Southern Ocean freshening during the last glacial

Chris S.M. Turney [1,2,3], Richard T. Jones[4], Steven J. Phipps [2,5], Zoë Thomas [1,2,3], Alan Hogg[6], A. Peter Kershaw[7], Christopher J. Fogwill [1,2], Jonathan Palmer [1,2,3], Christopher Bronk Ramsey [8], Florian Adolphi [9,10], Raimund Muscheler[9], Konrad A. Hughen[11], Richard A. Staff [8,12], Mark Grosvenor [4], Nicholas R. Golledge [13,14], Sune Olander Rasmussen [15], David K. Hutchinson [16], Simon Haberle[17], Andrew Lorrey[18], Gretel Boswijk [19] & Alan Cooper [20]

Contrasting Greenland and Antarctic temperatures during the last glacial period (115,000 to 11,650 years ago) are thought to have been driven by imbalances in the rates of formation of North Atlantic and Antarctic Deep Water (the 'bipolar seesaw'). Here we exploit a bidecadally resolved $^{14}$C data set obtained from New Zealand kauri (*Agathis australis*) to undertake high-precision alignment of key climate data sets spanning iceberg-rafted debris event Heinrich 3 and Greenland Interstadial (GI) 5.1 in the North Atlantic (~30,400 to 28,400 years ago). We observe no divergence between the kauri and Atlantic marine sediment $^{14}$C data sets, implying limited changes in deep water formation. However, a Southern Ocean (Atlantic-sector) iceberg rafted debris event appears to have occurred synchronously with GI-5.1 warming and decreased precipitation over the western equatorial Pacific and Atlantic. An ensemble of transient meltwater simulations shows that Antarctic-sourced salinity anomalies can generate climate changes that are propagated globally via an atmospheric Rossby wave train.

[1] Palaeontology, Geobiology and Earth Archives Research Centre, School of Biological, Earth and Environmental Sciences, University of New South Wales, Sydney, NSW 2052, Australia. [2] Climate Change Research Centre, School of Biological, Earth and Environmental Sciences, University of New South Wales, Sydney, NSW 2052, Australia. [3] ARC Centre of Excellence in Australian Biodiversity and Heritage, School of Biological, Earth and Environmental Sciences, University of New South Wales, Sydney, NSW 2052, Australia. [4] Department of Geography, University of Exeter, Exeter, Devon EX4 4RJ, UK. [5] Institute for Marine and Antarctic Studies, University of Tasmania, Private Bag 129, Hobart, TAS 7001, Australia. [6] Waikato Radiocarbon Laboratory, University of Waikato, Private Bag 3105, Hamilton 3216, New Zealand. [7] School of Earth, Atmosphere and Environment, Monash University, VIC 3800, Australia. [8] Research Laboratory for Archaeology and the History of Art, University of Oxford, Dyson Perrins Building, South Parks Road, Oxford OX1 3QY, UK. [9] Department of Geology—Quaternary Sciences, Lund University, 22362 Lund, Sweden. [10] Climate and Environmental Physics, University of Bern, CH-3012 Bern, Switzerland. [11] Woods Hole Oceanographic Institution, Woods Hole, MA 02543, USA. [12] Scottish Universities Environmental Research Centre (SUERC), University of Glasgow, Rankine Avenue, East Kilbride G75 0QF, UK. [13] Antarctic Research Centre, Victoria University of Wellington, Wellington 6140, New Zealand. [14] GNS Science, Lower Hutt 5011, New Zealand. [15] Centre for Ice and Climate, Niels Bohr Institute, University of Copenhagen, Juliane Maries Vej 30, 2100 Copenhagen, Denmark. [16] Bolin Centre for Climate Research and Department of Geological Sciences, Stockholm University, 10691 Stockholm, Sweden. [17] Department of Archaeology and Natural History and ARC Centre of Excellence in Australian Biodiversity and Heritage, College of Asia and the Pacific, Australian National University, Canberra, ACT 2601, Australia. [18] National Institute of Water and Atmospheric Research Ltd, Auckland 1010, New Zealand. [19] School of Environment, The University of Auckland, Private Bag 92019, Auckland 1142, New Zealand. [20] Australian Centre for Ancient DNA and ARC Centre of Excellence in Australian Biodiversity and Heritage, School of Biological Sciences, The University of Adelaide, Adelaide, SA 5005, Australia. Correspondence and requests for materials should be addressed to C.S.M.T. (email: c.turney@unsw.edu.au)

uring the last glacial, the global redistribution of heat is widely considered to be the cause of contrasting temperature trends between the hemispheres via an ocean pathway that operates on centennial timescales (the bipolar seesaw)[1–7], with abrupt cooling identified in the Greenland ice cores (Greenland Stadials; GS) leading the initiation of warming over Antarctica (Antarctic Isotope Maxima; AIM)[4, 6]. Although oceanic meridional heat transport variations are recorded through most of this period, recent work on North Atlantic marine sediments has highlighted limited changes in ocean circulation

between 33 and 18 kyr BP[8–10], and yet a pervasive antiphase temperature relationship continues to be observed between Greenland and Antarctica[4, 6], implying that additional mechanism (s) may have also operated during the glacial. Whilst atmospheric circulation variability may help to generate anti-phase hemispheric temperatures on monthly to decadal timescales—via latitudinal migration of the Intertropical Convergence Zone (ITCZ) and Southern Hemisphere storm tracks[6, 11]—the relationship is not consistent during all events[12]. To test whether other mechanisms in addition to the bipolar seesaw may have also

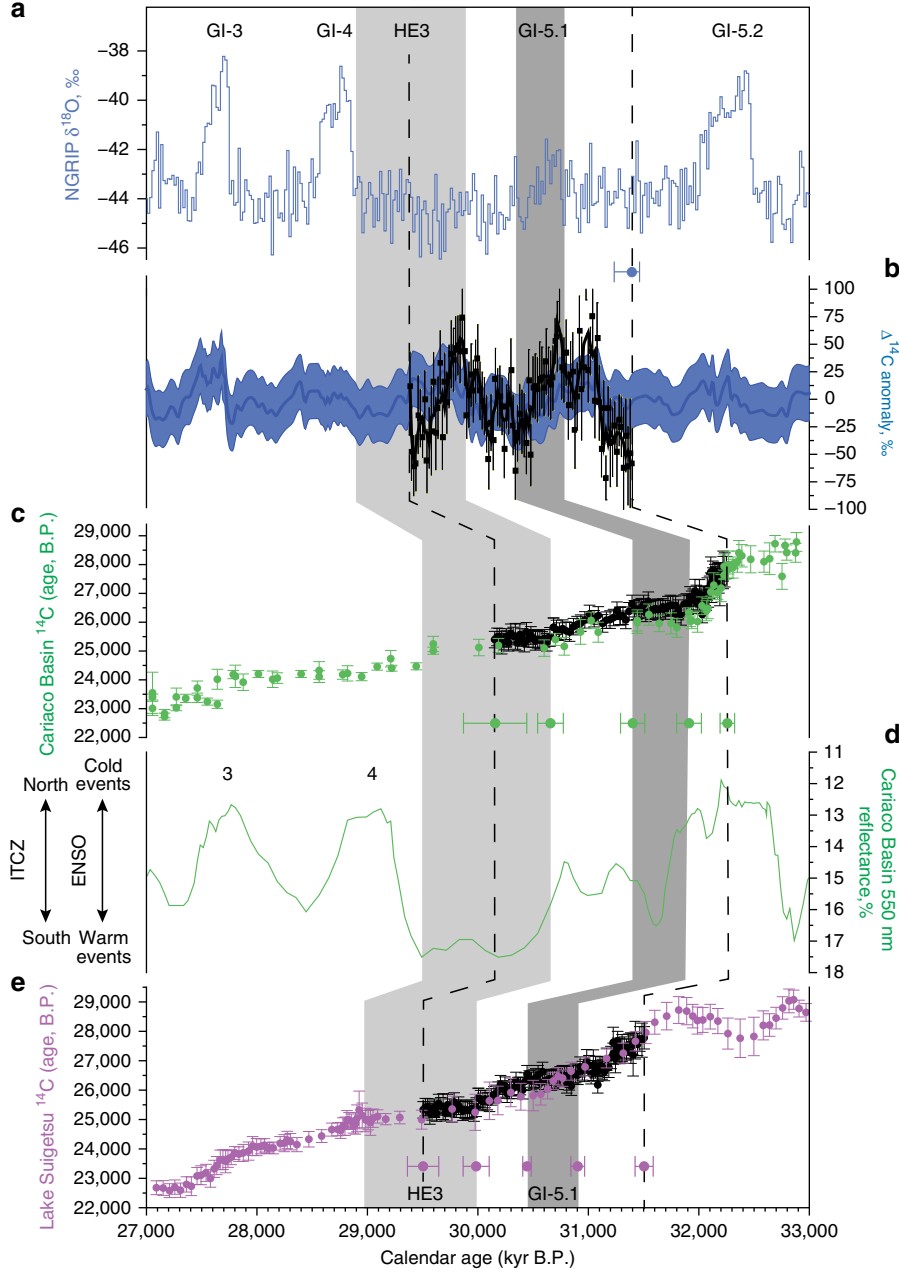

**Fig. 1** High-precision alignment of 2000-year long Finlayson 8 kauri tree [14]C measurements to key datasets. Bidecadal $\Delta^{14}$C data through the kauri tree (*black filled circles* with 3 point-running mean) compared to the Greenland (*blue*) $\delta^{18}$O and detrended [10]Be flux-derived $\Delta^{14}$C anomalies[10, 16] **a**, **b**. Comparison of kauri [14]C age measurements and Cariaco Basin[14] (**c** and **d**; *green symbols*) and Lake Suigetsu[13] (**e**; *purple symbols*) radiocarbon ([14]C) calibration data sets on the modelled Hulu Cave and SG06$_{2012}$ timescales, respectively. Green reflectance changes preserved in the tropical Atlantic Cariaco Basin marine sequence provide a measure of the meridional migration of the Intertropical Convergence Zone (ITCZ; **d**). Greenland Interstadials are numbered; Heinrich Event 3 (HE3, based on the climatostratigraphic position in MD95-2040)[17] and Greenland Interstadial 5.1 (GI-5.1) are shown by *light* and *dark* grey columns, respectively. The *dashed lines* denote the time range captured by the 2000-year kauri [14]C record used in this study[10]. Calendar age uncertainties for synchronization of records shown at 2$\sigma$ (95% confidence limits)

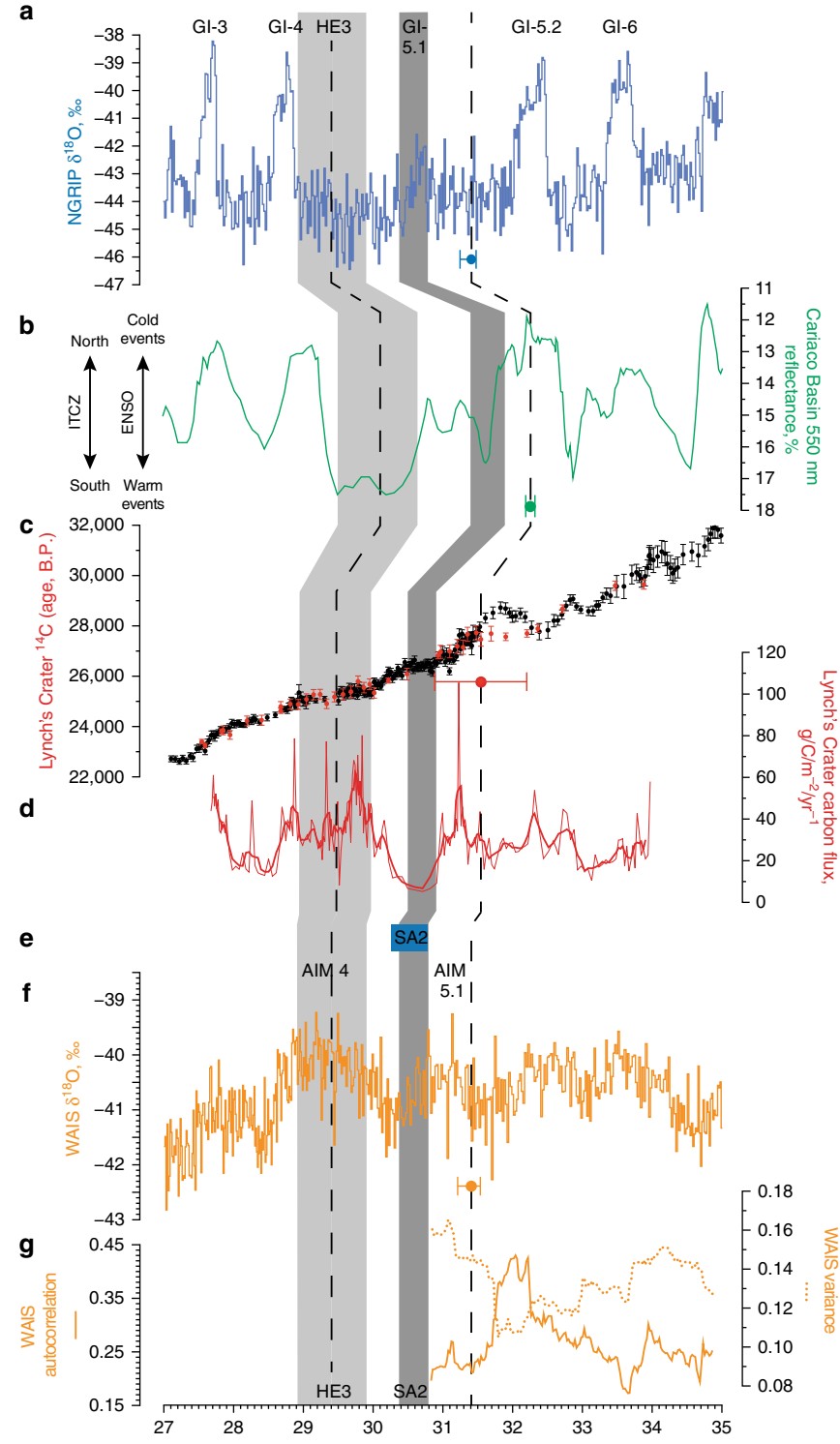

**Fig. 2** Global radiocarbon and environmental changes between 35 and 27 kyr BP. Comparison between North Greenland δ[18]O on the GICC05 timescale **a**[16], Cariaco Basin greenscale on the modelled Hulu Cave timescale **b**[14, 33], Bayesian wiggle-match of the Lynch's Crater (*red*) against the kauri-Lake Suigetsu calibration timescale (*black*)[10]; [14]C data sets with 1σ uncertainties (68% confidence limits) **c**, Lynch's Crater carbon flux (*red line*: 5-point running mean) **d**; the climatostratigraphic placement of South Atlantic ice rafted debris layer 2 (SA2)[21] **e**; and the West Antarctic Ice Sheet Divide (WAIS) δ[18]O (ref. [6]) on the WD2014(sync) (synchronized to GICC05) timescale (see Methods) **f** with calculated autocorrelation and variance values **g**. Greenland interstadials (GI) are numbered above the NGRIP δ[18]O record (**a**). The positions of Heinrich Event 3 (HE3), Greenland Interstadial 5.1 (GI-5.1) and SA2 in each record are shown as *grey columns*. The *dashed lines* denote the time range captured by the 2000-year kauri [14]C record used in this study[10]. Calendar age uncertainties for synchronization of records shown at 2σ

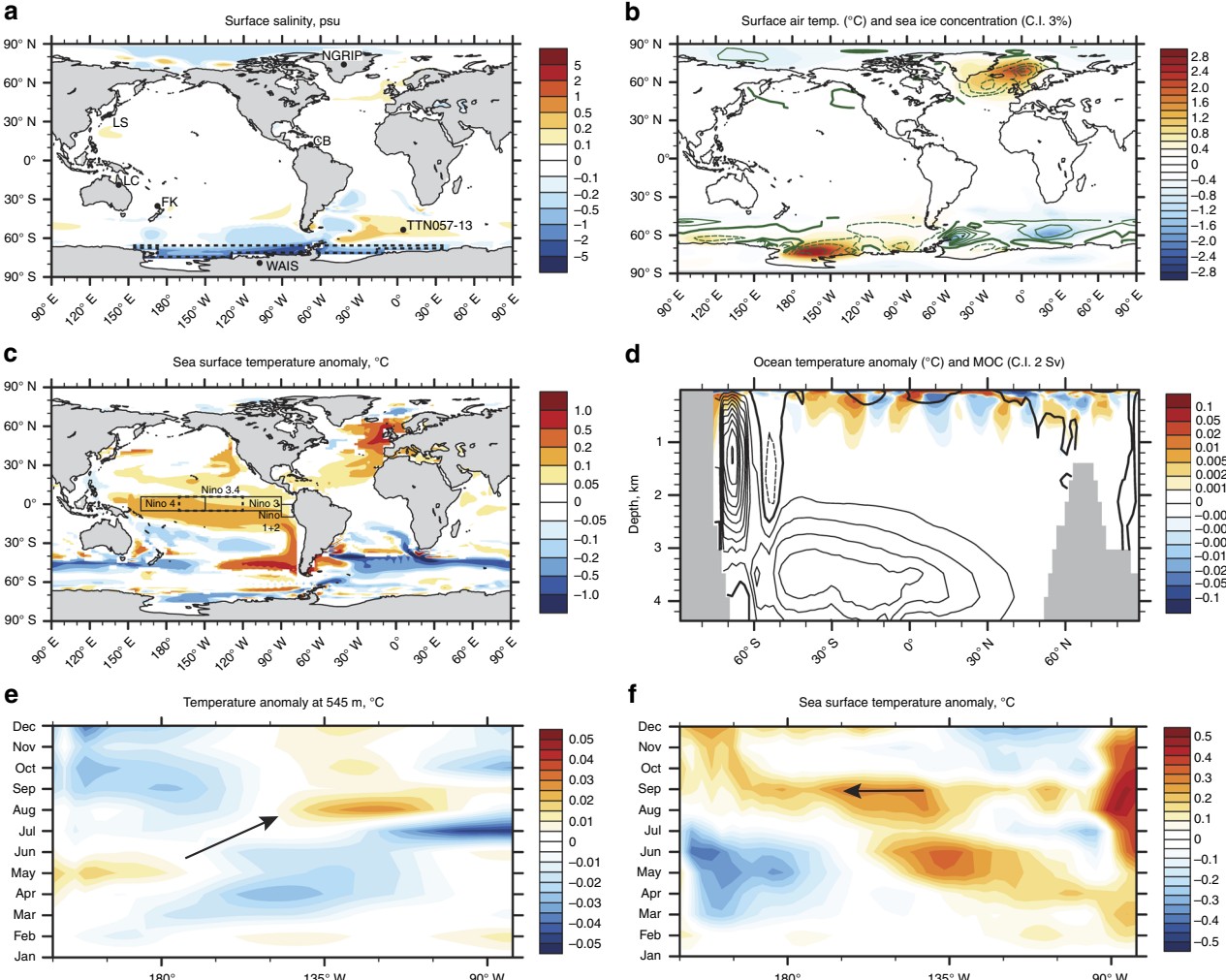

**Fig. 3** Summary of CSIRO Mk3L ensemble simulations showing the impact of a 338-year duration freshwater flux of 0.54 Sv into the Weddell and Ross Seas. Salinity anomaly is shown in **a** (*dashed lines* denote regions where freshwater applied with key site locations discussed in text shown). Surface air temperature (*colour*) and sea ice concentration anomalies (*green lines; solid* = positive, *dashed* = negative, with a contour interval of 3%) seen in **b** are not well-correlated with SST anomalies **c**, but sea ice concentration increases are highly correlated with salinity decreases in the Ross and Amundsen Sea sectors from the freshwater hosing. Global Meridional Overturning Circulation (MOC) anomaly is shown along with ocean temperature anomalies in **d**, where positive contours are solid, negative contours are dashed and the zero contour is emboldened, with a contour interval of 2 Sverdrups (Sv). The Southern positive cell represents reduced Antarctic Bottom Water (AABW) formation. Anomalies in **a–d** are averaged over the 338-year duration freshwater flux. Resulting seasonal equatorial (0°) Pacific eastward propagating Kelvin waves at the thermocline during the first year of freshwater application (545 m depth) in **e** and westward surface propagating Rossby waves in **f** identified by temperature changes. Significance *P* < 0.05

operated during the last glacial period requires the precise alignment of ice, marine and terrestrial records, something which hitherto has proved extremely challenging.

Here we exploit a newly-developed bidecadally resolved 2000-year long atmospheric $^{14}$C series obtained from a New Zealand sub-fossil kauri tree (*Agathis australis*)[10] to quantify time scale differences between key records of the last glacial and explore possible mechanisms to explain global climate patterns across this period.

## Results

**Alignment of palaeoclimate records.** Bayesian age modelling of the radiocarbon inflections and plateaux preserved in the New Zealand kauri against the tropical Atlantic marine Cariaco Basin and Japanese varved Lake Suigetsu (SG06$_{2012}$) records[13, 14] (see Methods) places the start of tree growth (and associated changes in atmospheric $^{14}$C content) on the timescale of these

sequences at 32,250 ± 70 BP and 31,510 ± 80 BP, respectively (Fig. 1). Note, unless $^{14}$C is stated, all ages reported here are calendar years relative to 1950 CE, Before Present (BP), with 2$\sigma$ uncertainty. We are able to splice the tree $^{14}$C record into the Lake Suigetsu sequence providing a refined atmospheric calibration curve and a calendar timescale for calculating radiocarbon concentration ($\Delta^{14}$C; ref. [10]). Cosmogenic radionuclides $^{14}$C and $^{10}$Be are produced via a nuclear cascade triggered when galactic cosmic rays collide with atmospheric atoms, with changes in the flux of cosmic rays leading to increased (decreased) $^{10}$Be and $^{14}$C production rates during times of low (high) solar activity and/or geomagnetic field intensity[15]. This offers us a unique opportunity to precisely align the kauri $^{14}$C variability with ice core $^{10}$Be on the Greenland Ice Core Chronology 2005 (GICC05)[10]. To compare the cosmogenic radionuclides, we modelled $\Delta^{14}$C from Greenland $^{10}$Be fluxes using a box-diffusion carbon cycle model run under preindustrial conditions[15]. We place the start of the kauri tree-ring series at 31,400$^{+70}_{-160}$ GICC05 BP, covering a

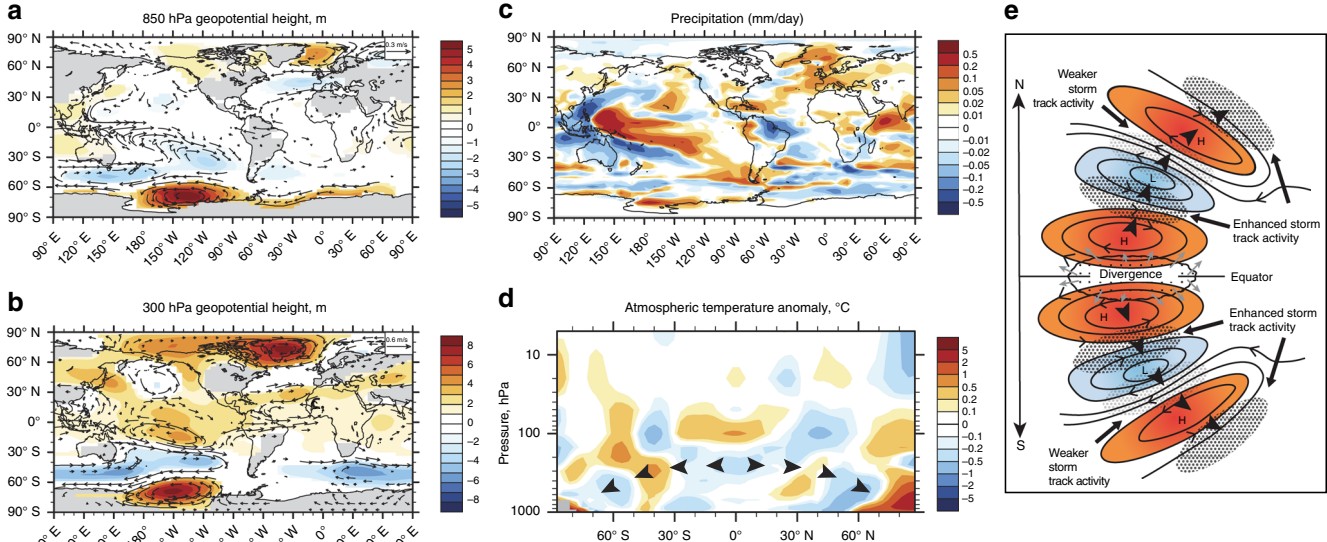

**Fig. 4** Modelled global atmospheric propagation of a Southern Ocean freshwater flux during the last glacial period. Geopotential height and wind anomalies at 850 hPa **a** 300 hPa **b** and global annual rainfall anomalies **c**. Zonally averaged global temperature anomalies for the atmosphere **d** reflect the characteristic pattern of westerly propagating Rossby waves. **a–d** produced from CSIRO Mk3L ensemble simulations of a 338-year duration freshwater flux of 0.54 Sv into the Weddell and Ross Seas. The schematic in **e** shows an idealized extratropical Rossby wave train (*solid black arrows*) associated with low (*blue*) and high (*red*) pressure systems generated by anomalous equatorial Pacific upper-level divergence[27, 28]. Significance $P < 0.05$

period of time that experienced warming associated with GI-5.1 and North Atlantic iceberg rafted debris layer Heinrich event 3 (IRD HE3; refs [16], [17]; Fig. 1 and Supplementary Fig. 1).

To compare global climate changes across the period defined by the kauri we also precisely aligned records from the equatorial west Pacific and Antarctica. We synchronised the West Antarctic Ice Sheet (WAIS) ice core $\delta^{18}O$ record to Greenland (creating a new timescale: WD2014$_{sync}$, synchronized at GI onsets to GICC05; see Methods). Incorporating the uncertainty associated with time-scale alignment we determine the start of the kauri $^{14}C$ series in WAIS to be 31,400$^{+140}_{-200}$ BP. We also undertook high-resolution analysis of swamp peat sediments preserved in Lynch's Crater, northeastern Australia, a region highly sensitive to changes in moisture-bearing equatorial southeast trade winds in the western equatorial Pacific (Supplementary Fig. 2)[18]. Bayesian age modelling of 44 $^{14}C$ ages from the Lynch's Crater sequence allowed us to align it against the combined kauri-Lake Suigetsu radiocarbon curve. We place the onset of the 2000-year period in Lynch's Crater at 31,540 ± 660 BP (on the SG06$_{2012}$ timescale).

## Discussion

The alignment of $^{14}C$ and $^{10}Be$ records allows us to explore global atmospheric and ocean teleconnections across a period of pronounced abrupt climate change (Fig. 2). Here we identify a ~460-year-long radiocarbon plateau at ~26.4 $^{14}C$ kyr BP coincident with GI-5.1 and a ~480-year duration ~25.4 $^{14}C$ kyr BP plateau during a period of enhanced delivery of freshwater and debris-laden icebergs into the North Atlantic associated with HE3 (refs [8], [17], [19]; Fig. 1 and Supplementary Fig. 1). The parallel changes in Cariaco Basin surface water (with a mean reservoir age of 372 ± 35 $^{14}C$ years) and atmospheric $^{14}C$ levels argue against a stratification of surface waters[20] and collapse of the Meridional Overturning Circulation (MOC)[8–10] implying the ocean circulation was not appreciably shutdown during the 2000 years captured by our tree-ring series (including during HE3; Fig. 1). Of importance, we find a southward migration of the ITCZ concurrent with the warmth of GI-5.1 (and sustained Antarctic

cooling between AIM5.1 and 4), opposite to that anticipated for North Atlantic warming[5] (Fig. 2).

To explore alternative global atmospheric teleconnections, we undertook multi-proxy analyses of the Lynch's Crater sediments. Contiguous carbon analysis allows us to reconstruct changing carbon flux across this period and identifies a ~420-year down-turn that commences at 30,920 ± 340 BP and coincides with the ~26.4 $^{14}C$ kyr BP radiocarbon plateau (Fig. 2). This event is associated with low sediment accumulation rates and low sedge to grass ratios[18] that both imply relatively dry conditions (Supplementary Fig. 3). We consider this local hydroclimatic response to be a consequence of weakened moisture-laden equatorial Pacific trade winds[18] which appear to have been synchronous (within the precision of the chronologies reported here) with sustained southward migration of the ITCZ.

Tipping point analysis of the WAIS $\delta^{18}O$ (WD2014$_{sync}$) and the Lynch's Crater sequences shows no evidence of critical slowing down in the approach to cooling over the West Antarctic and drying in tropical Australia (which would be manifested as an increase in autocorrelation and variance on the approach to a tipping point) (Fig. 2, and Supplementary Figs. 4 and 5), suggesting that a rapid external climate trigger is more likely to be responsible than a birfucation associated with slow internal forcing. Although there is no evidence for a weakening of MOC across the period of the radiocarbon plateau at ~26.4 $^{14}C$ kyr BP (ref. [7–10]; Fig. 1), a substantial increase in Antarctic ice-sheet discharge occurred immediately prior to the onset of AIM4 evidenced by South Atlantic IRD layer 2 (SA2), south of the Polar Front in South Atlantic marine sequences (e.g., TTN057-13)[21] (Fig. 2 and Supplementary Fig. 6; see Methods).

To investigate whether a sustained Antarctic ice-sheet drawdown and associated freshwater pulse into the Southern Ocean might explain the observed pattern of global changes we undertook an ensemble of transient meltwater experiments using the fully coupled CSIRO Mk3L global climate system model[22]. For SA2 we applied freshwater fluxes of 0.54 and 0.27 Sv to the Ross and Weddell Seas for 338 years, comparable to the magnitude and duration of the Last Termination Meltwater Pulse

1A (MWP-1A)[23] (see Methods). Following a flux of 0.54 Sv freshwater, all simulations show surface atmospheric cooling over large parts of the Antarctic continent, with significant surface warming (> 1.5 °C annual mean temperature increase) and decreases (increases) in sea ice extent in the Ross Sea and North Atlantic (South Atlantic) accompanied by pervasive warming in the equatorial Pacific (Fig. 3). Although we find a reduction in the simulated rate of Antarctic Bottom Water (AABW) formation, arising from vertical stratification of the water column in the Ross and Weddell Seas, there is no change in the MOC during our hosing simulations.

Previous modelling studies have implied that Southern Ocean salinity changes can lead to an equatorial and global climate response within a year (refs. [24–26]). A key part of this rapid ocean mechanism is the generation of barotropic Kelvin waves that propagate along the Antarctic coastline and are directed north along the western margin of the Pacific Ocean via the South Pacific Rise. The result is excitation of equatorial baroclinic Kelvin waves that leads to pronounced surface warming[25]. Within four months of freshwater hosing in the Ross and Weddell Seas, subsurface temperature anomalies representing eastward travelling baroclinic Kelvin waves appear in our simulations along the equatorial thermocline, crossing the Pacific in approximately three months (Fig. 3e). The arrival of these Kelvin waves in the east Pacific triggers highly non-linear ocean-air interactions through the reflection of Rossby waves from the eastern boundary and generation of anomalous westerly equatorial airflow. These waves increase sea surface temperatures (> 0.4 °C), comparable to previous studies[24–26] (Fig. 3f).

The model simulations indicate an acute decrease in geopotential height across the mid-latitudes but an increase in the Ross Sea and North Atlantic, the latter coinciding with the location of maximum surface warming (Fig. 4). The warmer equatorial SSTs are associated with deep convection and upper-level divergent flow at 300 hPa, forcing what appears to be an atmospheric Rossby wave train in the extratropics, manifested as low pressure anomalies across the mid-latitudes that extend polewards as high pressure anomalies[27, 28] (Fig. 4b). In parallel with these changes we also find marked shifts in equatorial and extra-tropical precipitation with drier conditions in the western Pacific (including northeast Australia) and northern South America, accompanied by wetter conditions over the central Pacific (Fig. 4c). The simulated spatial SST and the precipitation anomaly fields closely resemble the modern (post-1979) global El Niño-like pattern of rainfall change[29] (Fig. 4 and Supplementary Fig. 8). The increase in Cariaco Basin greenscale that is often interpreted as a measure of southward migration of the ITCZ in response to North Atlantic cooling is consistent with central equatorial Pacific warming associated with El Niño warming, with drier conditions over northern South America[30] (Fig. 2). The eastward displacement of the convective loci and northeast movement of the South Pacific Convergence Zone seen in the model results is also associated with El Niño events[29], and is highly relevant to the observed drying at Lynch's Crater. The similarity between proxy time series and 100-year simulation time slices for temperature and precipitation anomalies add further support to this mechanism, with peak warming at the onset of SA2 but persistent drying through the period of freshwater hosing (Supplementary Figs. 9 and 10). Although the impacts of a flux of 0.27 Sv in the Southern Ocean are smaller than the impacts of a flux of 0.54 Sv, the same trends are observed globally (Supplementary Figs. 11–14), suggesting that this mechanism can operate across a range of freshwater hosing applications.

The similar pattern of observed and modelled temperature, precipitation and sea ice anomalies suggests that coupled Antarctic-Southern Ocean dynamics contributed to some global

events through rapid ocean-atmospheric teleconnections. Whilst our findings do not preclude the operation of an ocean bipolar see-saw during other periods within the last glacial, these results provide an additional mechanism for driving past (and future) global change. Although our data and model simulations suggest a remarkably rapid ocean-atmosphere response to Southern Ocean freshening, the relatively muted expression of GI-5.1 implies that this forcing may result in smaller magnitude changes to those driven by the bipolar seesaw. Our findings therefore suggest that contrasting high-latitude temperature trends during the last glacial can also be driven from Antarctica and the Southern Ocean via ocean-atmosphere teleconnections.

## Methods

**High-precision alignment of terrestrial and ice records.** Multiple trees of different ages across the late Pleistocene were extracted from a swamp on Finlayson Farm (35° 83′ S, 173°64′ E) in 1998 (refs [31, 32]) from which we have identified a 2000-year long kauri log (henceforth 'Finlayson 8') to generate a bidecadally resolved [14]C series to reconstruct atmospheric radiocarbon content[10]. To elucidate changes in atmosphere and ocean circulation across HE3 and GI-5.1, we exploited temporal signals (e.g., using so-called 'radiocarbon plateaux') in two high-resolution radiocarbon datasets that allow high-precision alignment of the series: a terrestrial record of atmospheric [14]C generated from the Japanese varved Lake Suigetsu sequence (using the SG06$_{2012}$ timescale)[13], and the Venezuelan Cariaco Basin sedimentary sequence, which preserves a record of surface water [14]C and changes in the Intertropical Convergence Zone (ITCZ) that parallel Greenland Interstadial events[12, 14, 33]. The [14]C series was calibrated using a Poisson process deposition model (P_Sequence)[34] with the General Outlier analysis option[35] in OxCal 4.2. Using Bayes theorem, OxCal identifies possible age solutions for the kauri series of [14]C ages in the Cariaco Basin and Lake Suigetsu calibration data sets. To accommodate a possible collapse in the marine reservoir age (ΔR) through the Cariaco Basin sequence (420 [14]C years), a Delta_R (ΔR) with the prior U(0,420) was used, allowing [14]C measurements to assume atmospheric values if required. Whilst neither of the [14]C calibration datasets applied here are bidecadally resolved across the period spanning HE3 and GI-5.1, there is sufficient structure in both series to precisely align the Finlayson 8 [14]C data set, providing a chronology for the kauri, and in the case of the Cariaco Basin, demonstrating no significant sustained changes in marine reservoir ages across this period[10].

To compare the kauri [14]C series with the Greenland Ice Core Chronology 2005 (GICC05)[16, 36, 37], we aligned variations in atmospheric radiocarbon concentration (Δ[14]C) as recorded by the New Zealand tree against the [10]Be measurements from the GRIP ice core[10, 15, 38, 39]. We placed the kauri Δ[14]C sequence on the GICC05 chronology using previously reported methods[10, 40]. Due to carbon cycle effects the atmospheric Δ[14]C signal was dampened and delayed compared to [14]C production rate variations. Hence, we modeled Δ[14]C from GRIP [10]Be fluxes assuming these are proportional to global production rate variations using a box-diffusion carbon cycle model run under preindustrial conditions[15, 41]. The long-term trends in Δ[14]C can be increasingly affected by carbon cycle changes which are not reflected in [10]Be and difficult to quantify[15, 40]. Due to the radioactive decay of [14]C and the relatively old ages of the kauri, [14]C measurement uncertainties are too large to resolve the centennial Δ[14]C variations that have been used for [10]Be/[14]C synchronization by Adolphi and Muscheler (2016) during a more recent period. Hence, we high-pass filtered the [10]Be-based Δ[14]C record with a cutoff frequency of 1/2000 per year and linearly detrended the kauri Δ[14]C data. Given the length of the kauri chronology of 2000 years, this detrending is comparable to a 2000-year high pass filter but it avoids edge-effects induced by filtering. It is difficult to estimate an uncertainty to the GRIP [10]Be-based Δ[14]C record. Uncertainties arising from the unknown history of the carbon cycle as shown by Muscheler et al.[15] may be systematic (on the time scales considered here), and would be removed by filtering the modelled Δ[14]C. On the other hand, a different state of the ocean's deep convection and/or air–sea gas exchange rates would impact on the amplitude of Δ[14]C variations for a given [14]C production rate change[42], which would also be present after filtering. We therefore assigned a 1σ uncertainty of 25‰ to the GRIP [10]Be-based Δ[14]C record. This is consistent with simulated millennial Δ[14]C variations that can be induced by carbon cycle changes alone[42]. It should be noted, however, that the derived fits of the kauri sequence onto the modelled Δ[14]C record depend very little in practice on the assumed error for the GRIP [10]Be-based record. Details regarding the mathematical formulation of the method and its application to [14]C and [10]Be records have been published elsewhere[40, 43]. To take into account the relative uncertainty of the ice core time scale we repeated the calculations assuming linear stretches from −100 to 100% of the GICC05 maximum counting error in steps of 10% for the period of overlap.

**The Cariaco Basin marine record.** The Venezuelan Cariaco Basin marine palaeoclimate record is the 550 nm reflectance data from ODP Hole 1002C. The age model is based on matching abrupt palaeoclimate shifts between the Cariaco Basin and Hulu Cave speleothem records. The age model is an update on previous

studies[10, 14, 33] using more highly-resolved Hulu $\delta^{18}O$ data that allow more precise identification of the timing of Dansgaard-Oeschger (D-O) events used as tie-points. Regardless, the conclusions of this study are insensitive to the chronology used given the green reflectance and $^{14}C$ data are directly linked. Essentially, these are the palaeoclimate data that accompany the updated Cariaco Basin marine $^{14}C$ ages used in IntCal13[44].

**Ice rafted debris events in marine sediments.** The IRD (HE3) and summer SST records from MD95-2040[17] were placed on a calibrated timescale using a Poisson process deposition model (P_Sequence)[34] with the General Outlier analysis option[35] in OxCal 4.2. Radiocarbon ages were calibrated using the Marine13 calibration data set[44] and Greenland isotope event boundaries[16] were imported. To accommodate uncertainties in the marine reservoir age, a Delta_R with the prior U (-400,0) was used (Supplementary Table 1). A mean $\Delta^{14}C$ of $278 \pm 90$ years was determined for the sequence (Supplementary Fig. 1). The IRD and planktic $\delta^{18}O$ data from the South Atlantic (SA2) are based on reported calibrated ages (Supplementary Fig. 6)[21]. Whilst previous work has suggested the IRD is dominated by volcanic ash shard fallout from the South Sandwich Island onto sea ice in the South Atlantic[45], the parallel increase in quartz grains within this horizon remains consistent with an increased discharge of Antarctic ice[46].

**Lynch's Crater site description.** To reconstruct western equatorial Pacific changes across the period spanning HE3 and GI-5.1 we investigated the tropical swamp sequence at Lynch's Crater, northeast Queensland on the Atherton Tableland (17.37°S, 145.69°E; Supplementary Fig. 2)[18, 47–49], a site highly sensitive to precipitation changes delivered by moisture-bearing southeast trade winds[18]. The volcanic crater contains around 64 m of lake and peat sediments, and records substantial sedimentological and palaeoecological change in response to climatic oscillations as well as more sustained changes in plant taxon and community distributions.

**Lynch's Crater radiocarbon dating and climate reconstruction.** The original Lynch's Crater chronology was based upon bulk radiocarbon ($^{14}C$) ages derived from the uppermost 16 m of the sequence, assuming linear accumulation and taking into account the moisture content of the peat[50]. Using a Livingstone cored sequence obtained in 1998, we undertook comprehensive $^{14}C$ sampling (44 peat samples) between 4 and 7 m, with most of the ages focused on 4.9 to 6.95 m (spanning the period of accumulation that includes SA2). Bulk samples were pretreated using an acid-base-acid (ABA) protocol (with multiple base extractions) and then combusted and graphitized in the University of Waikato AMS laboratory, with $^{14}C/^{12}C$ measurement by the University of California at Irvine (UCI) on a NEC compact (1.5SDH) AMS system. The pretreated samples were converted to $CO_2$ by combustion in sealed pre-baked quartz tubes, containing Cu and Ag wire. The $CO_2$ was then converted to graphite using $H_2$ and a Fe catalyst, and loaded into aluminium target holders for measurement at UCI. Samples of ABA-pretreated Waikato OIS7 kauri background standard[51] were prepared and measured with the unknown age samples. Results on the OIS7 blank ranged from 0.0008 to 0.0015 $F^{14}C$ (57.2–52.4 kyr BP) with a mean of 0.0012 $F^{14}C$ (53.8 kyr BP), with an assumed uncertainty of $\pm 30\%$. The quoted uncertainties are compiled from uncertainties in the Background and Modern standards, as well as from the variability in the repeated runs on each sample and from counting statistics. The radiocarbon ages were calibrated against the new combined kauri-Lake Suigetsu calibration dataset[10] using a P_sequence deposition model with the General Outlier analysis option in OxCal 4.2[34, 35] (Supplementary Software 1). The resulting age model generated an Agreement Index of 170.2% ($A_{overall} = 161.4$; Supplementary Table 2), exceeding the recommended > 60%[52].

In parallel with the comprehensive dating, we undertook high-resolution Total Organic Carbon (%TOC) analysis of the peat sediments straddling the time of SA2. Contiguous 1 cm samples were taken between 5 and 7 m and measured for TOC, determined using a LECO TruSpec CN analyser at the University of New South Wales Analytical Centre following standard techniques[53]. The high-resolution dating allowed us to reconstruct changes in the carbon flux (the amount of carbon sequestered) at Lynch's Crater using previously reported methods[54]. The sustained decrease in carbon flux and associated drying is consistent with other records elsewhere on the Atherton Tableland, including a contemporaneous layer of gypsum ($CaSO_4$) representing a significant moisture deficit in Strenekoff's Crater[55], approximately 5 km from Lynch's Crater.

**West Antarctic ice sheet alignment.** The original West Antarctic Ice Sheet (WAIS) Divide deep ice core WD2014 chronology was constructed through a combination of annual-layer counting (0-31.2 kyr BP) and interpolar methane synchronisation (31.2-68 kyr BP)[56, 57]. The period of interest here lies exactly in this transition zone. Therefore, we used an alternative WAIS Divide chronology for the 2700–2850 m depth range (27–31.2 kyr BP), called WD2014$_{sync}$, in which the methane synchronization method[56] is extended to include DO events 3 through 5.1 (Supplementary Data 1). Note that in the original chronology this depth range was dated using annual-layer counting. The methods and uncertainty estimation in the new WD2014$_{sync}$ chronology follow those described in Buizert et al[56].

**Tipping point analysis.** Tipping point analysis was undertaken to test whether changes in the Lynch's Crater carbon sequestration rate and the West Antarctic Ice Sheet (WAIS) Divide $\delta^{18}O$ (WD2014$_{sync}$) might have been a result of long-term forcing. Here statistical properties of the data were analysed to see whether signals of 'critical slowing down' can be detected[58–60]. Critical slowing down occurs when a system approaches a bifurcation, characterised by a decreasing recovery rate to small perturbations. This can be detected as a short-term increase in the lag-1 autocorrelation of the time series[61], often also accompanied by an increase in variance[59]. However, bifurcational tipping is just one subset in the group of tipping points[62]; noise-induced tipping points are not caused by a slow internal forcing, and are rather forced by an external trigger. The detection of critical slowing down in the case of a noise-induced tipping would therefore not be expected. To test for the presence of critical slowing down, data pre-processing was necessary before the autocorrelation and variance could be measured. This included detrending to remove long-term trends using the Gaussian kernel smoothing function, applied over a suitable smoothing bandwidth, and interpolation to provide equidistant data points. Autocorrelation and variance were then measured over a sliding window[59] on the resulting residual data after smoothing. The Kendall tau rank correlation coefficient was then applied to provide a quantitative measure of the trend. This metric varies between + 1 and −1, where higher values indicate a stronger increasing trend due to a greater concordance of pairs. Since there are trade-offs in the choice of size of smoothing bandwidth and sliding window of analysis, a sensitivity analysis was undertaken by varying the size of these parameters. The Kendall tau values are then displayed in a contour plot to show how the different parameter choices affect the analysis. Consistent Kendall tau values in the contour plot indicate that the analysis is not sensitive to the parameter choices.

Similar results to the WAIS $\delta^{18}O$ record[6] (WD2014$_{sync}$) were obtained for tipping point analysis of the carbon accumulation data from Lynch's Crater during the shift to drier conditions in tropical Australia (Fig. 2, and Supplementary Figs. 4 and 5). A slight increasing trend in variance is found in the WAIS data (though not the Lynch's Crater record), which may suggest that the system was becoming noisier. However, it must be acknowledged that due to the challenges in identifying false positives and false negatives[60], particularly in sparsely sampled palaeoclimate data[60], these results cannot be considered conclusive, but are suggestive of a lack of bifurcation, consistent with a single forced event.

**Modelling the impact of a Southern Ocean freshwater pulse.** Our simulations used the Commonwealth Scientific and Industrial Research Organisation Mark version 3L (CSIRO Mk3L) climate system model version 1.2, comprising fully interactive ocean, atmosphere, land and sea ice sub-models[22, 63, 64]. CSIRO Mk3L employs a reduced horizontal resolution and is designed for millennial-scale climate simulations. The ocean model is a 'rigid lid' model (i.e., the surface is fixed), with a horizontal resolution of 1.6° latitude × 2.8° longitude and 21 vertical levels, while the atmosphere model has a horizontal resolution of 3.2° latitude × 5.6° longitude and 18 vertical levels. A smoothed version of the 2-Minute Gridded Global Relief Data (ETOPO2v2) topography (https://www.ngdc.noaa.gov/mgg/global/etopo2.html) was used for the simulations (Supplementary Fig. 7). Despite the reduced horizontal resolution, the model has a stable and realistic control climatology and has demonstrated utility for simulating the past, present and future evolution of the climate system[63–65].

In this study, an ensemble modelling approach was employed in which each experiment was repeated three times. Three experiments were conducted. In the first experiment, the model was used to conduct a transient simulation of the background climate during the period 32 to 28 kyr BP, in the absence of any catastrophic freshwater fluxes. Each of the three ensemble members was initialized from a different year of a pre-industrial control simulation. Transient changes in the Earth's orbital parameters and greenhouse gas concentrations were imposed (Supplementary Table 3). Land ice was fixed at its extent 21 kyr BP, using the ICE-5G version 1.2 reconstruction[66].

In the second and third experiments, we simulated a SA2-like event by introducing 0.54 Sv and 0.27 Sv of freshwater into the Weddell and Ross Seas over a period of 338 years, comparable to the duration of Meltwater Pulse 1A (MWP-1A) during the Last Termination[23, 67]. The 0.54 Sv flux is equivalent to an increase of 16m in global sea level, and was chosen assuming Antarctica was the single source of the upper limit of reconstructed changes in sea level rise across AIM4[68]. Thus, the second experiment was identical to the first, except for a 338-year duration freshwater flux of 0.54 Sv beginning at 29.5 kyr BP consistent with reconstructed glacial meltwater discharge for MWP-1A[23] and imposed over the Weddell and Ross Seas, locations indicated by the Parallel Ice Sheet Model (PISM) for this same event[69]. The third experiment was identical to the second, but used a 0.27 Sv flux to test whether the mechanism identified with the 0.54 Sv was also observed using a smaller freshwater hosing (Supplementary Figs. 11 and 12). 29.5 kyr BP was selected as the period for freshwater hosing application to minimize the effect of changing greenhouse gas forcing. Importantly, our results and those reported elsewhere suggest the magnitude of the forcing to be linearly scaled with freshwater forcing[24, 25] (see also Extended Data Fig. 4 in ref. [70]). Each ensemble member was branched off the equivalent ensemble member from the first experiment.

**Data availability**. All the new data are provided in Supplementary Information and have also been lodged with the NOAA/World Data Center for Paleoclimatology at https://www.ncdc.noaa.gov/paleo/study/22391.

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

## Acknowledgements

This work was funded by the Australian Research Council (FL100100195, DP170104665 and SR140300001) and the Natural Environment Research Council (NE/H009922/1 and NE/H007865/1). We thank Dr Charlotte Cook and Dr Sarah Kelloway for helping to process the Lynch's Crater samples, and Dr Christo Buizert for calculating the WD2014$_{sync}$ chronology.

## Author contributions

C.T. and R.J. conceived the research; C.T., R.J., S.P., Z.T., A.H., J.P., C.B.R., R.S., F.A., R.M. designed the methods and performed the analysis; C.T. wrote the paper with input from R.J., S.P., Z.T., A.H., P.K., C.J.F., J.P., C.B.R., F.A., R.M., K.H., R.S., M.G., N.G., S.R., D.H., S.H., A.L., G.B. and A.C.

## Additional information

**Competing interests:** The authors declare no competing financial interests.

