## [Peer Review File · Nature Communications]

Reviewers' comments:

Reviewer #2 (Remarks to the Author):

Review of the manuscript 'Rapid global ocean-atmosphere response to Southern Ocean freshening during the last glacial' by Turney et al.

General statement:

Some time ago I refereed the original version of the manuscript entitled "Rapid global ocean-atmosphere response to Southern Ocean freshening during the last glacial" by Turney et al. for [redacted].

In the new submission Turney et al. present a revised version of this manuscript. This version benefits from a more balanced handling of the general implications for first order DO-events and a model analysis plus interpretation that focuses more clearly on the dynamics of the abrupt global ocean-atmosphere responses. While these improvements are appreciated the integration of the model results into the data framework during SA2 and the corresponding sequence of events, especially in the high-latitudes, is still difficult to follow.

Major Comments:

The length of the SA2 event in the data is ~400 yrs (Fig. 2) and the simulated duration of the freshwater flux in the model experiments is 338 yrs, i.e. both have a similar length of time. In the Antarctic records the SA2 interval is indeed a period of sustained cooling, while the other records show a more complex temporal evolution within SA2. For instance in the Greenland record warming only takes place during less than the first half of SA2 (Fig. 2a), but the cooling signal thereafter (still within SA2) is of similar magnitude. This suggests that the simulated model behaviour addresses the contrasting high-latitude temperature trends during the first half of SA2. In other words: Are the freshwater experiments supposed to represent the full duration of SA2 or just the first half of SA2? This can't be answered by the presented model analysis that shows time-averaged output for the whole duration of the freshwater flux experiments.

A viable option to address this is to show the time evolution of simulated changes within the freshwater flux intervals by time-series in key regions and temporal changes using global maps. This is a critical requirement to have a clear allocation of the model results to the climate changes as recorded by data during SA2 (Fig. 2). Otherwise the significance of the presented mechanism with respect to high-latitude coupling is ambiguous.

Specific comments:

- Climate responses are known to be strongly dependent on the background climate and the CO₂ forcing during the full simulation interval (32 to 28 kyr BP) is quite large. In this study the freshwater experiments are based on a transient simulation of the background climate during the period 32 to 28 kyr BP. What is the reason to start the freshwater experiments at 29.5 kyr BP and not at the beginning of SA2 (~30.8 kyr BP)? This needs to be explained and discussed.

- It is difficult to read the following sentence:

'Following a flux of 0.54 Sv freshwater, all simulations show surface atmospheric cooling over large parts of the Antarctic continent, with significant surface warming (>1.5 °C annual mean temperature increase) and increases (decreases) in sea ice extent in the South Atlantic (Ross Sea and North Atlantic) accompanied by pervasive warming in the equatorial Pacific (Figure 3).'

Do you really mean a warming in the South Atlantic?

- Abstract: 'Using a fully coupled climate system model we undertake an ensemble of transient meltwater simulations and find that southern salinity anomalies can trigger low-latitude temperature changes that are atmospherically propagated globally via a Rossby wave train, with implications for past and future change.'

Please specify the implications.

General Recommendation:

The study is still an interesting contribution for a broader research community and I would be happy to have a look on a revised manuscript version.

Reviewer #3 (Remarks to the Author):

This is my third review of the paper by Turney et al. I have gone through the entire paper once again and read through their response to the reviewers. I have only minor comments for the paper, which the authors can decide to incorporate if they like, as they are not major. Overall, I think the authors have done an outstanding job both addressing and carefully considering all of the reviewer comments, and notably done so very cordially given the reviews (including my own), which asked quite a lot of them and required they do additional analyses and simulations. I personally appreciated this exchange.

Minor Comments:

Citations 1-5: Wouldn't the most appropriate reference for the bipolar seesaw be Stocker and Jansen, 2003; Broecker, 1998; and/or Stocker, 1998? These other papers build on the idea but the credit for the idea comes much earlier – in fact the first observations and discussion of the significance (without naming it) comes from the early work of Mix, Ruddiman, and McInyre 1986.

Lines 149-150: I'm confused about exactly what exact period the authors are referring. If they are referring to around H3, then I'd say: are you sure there isn't evidence of weakened overturning? The paper by Böhm et al. 2015 Nature (Figure 3) and the Henry et al. 2016 (Figure 2) papers both show a nice summary of overturning proxies and H3 does appear to be demonstrating some reduction.

Reviewer #2 (Remarks to the Author):

The length of the SA2 event in the data is ~400 yrs (Fig. 2) and the simulated duration of the freshwater flux in the model experiments is 338 yrs, i.e. both have a similar length of time. In the Antarctic records the SA2 interval is indeed a period of sustained cooling, while the other records show a more complex temporal evolution within SA2. For instance in the Greenland record warming only takes place during less than the first half of SA2 (Fig. 2a), but the cooling signal thereafter (still within SA2) is of similar magnitude. This suggests that the simulated model behaviour addresses the contrasting high-latitude temperature trends during the first half of SA2. In other words: Are the freshwater experiments supposed to represent the full duration of SA2 or just the first half of SA2? This can't be answered by the presented model analysis that shows time-averaged output for the whole duration of the freshwater flux experiments.

A viable option to address this is to show the time evolution of simulated changes within the freshwater flux intervals by time-series in key regions and temporal changes using global maps. This is a critical requirement to have a clear allocation of the model results to the climate changes as recorded by data during SA2 (Fig. 2). Otherwise the significance of the presented mechanism with respect to high-latitude coupling is ambiguous.

Not including time-evolving plots of temperature and precipitation was an oversight. We had the outputs and with hindsight should have included

them. We have now added four new supplementary figures (in the revised version given as Figs. S9, S10, S13 and S14). These figures show 100-year timeslices of the evolving global temperature and precipitation response to freshwater hosing in the Southern Ocean for both the 0.54 Sv (see below) and 0.27 Sv simulations.

Figure S9: Statistically-significant ensemble mean 100-year global surface air temperature anomalies using the CSIRO Mk3L climate system model^{6,7} during 0.54 Sv freshwater hosing (panels a-c) and the 100 years after the hosing ceases (d). Anomalies are calculated relative to equivalent transient simulations in which no freshwater hosing is applied. Significance $p_{field} < 0.05$.

Figure S10: Statistically-significant ensemble mean 100-year global precipitation anomalies (mm/day) using the CSIRO Mk3L climate system model^{6,7} during 0.54 Sv freshwater hosing (panels a-c) and the 100 years after the hosing ceases (d). Anomalies

As can be seen, our ensemble of simulations not only capture pervasive warming over the North Atlantic across the full period but also capture the detail preserved in the Greenland ice core: specifically, peak warming at the onset of SA2 and more muted warming during the latter part of the freshwater hosing. In contrast, the precipitation anomaly appears to strengthen through this period, implying the tropics continue to respond to southern freshwater forcing but the North Atlantic teleconnection weakens; something we intend to investigate in future studies but beyond the scope of the current study.

Specific comments:

- Climate responses are known to be strongly dependent on the background climate and the CO₂ forcing during the full simulation interval (32 to 28 kyr BP) is quite large. In this study the freshwater experiments are based on a transient simulation of the background climate during the period 32 to 28 kyr BP. What is the reason to start the freshwater experiments at 29.5 kyr BP and not at the beginning of SA2 (~30.8 kyr BP)? This needs to be explained and discussed.

We do sympathise with this view but it was important to isolate the impact of freshwater hosing in the Southern Ocean from changing forcing from changing greenhouse gas concentrations. As a result, we chose to force our model during a period of relatively complacent (unchanging) concentrations. If we had chosen the later period then we risked having highly variable greenhouse gas levels and/or risked conflating a delayed response of the Earth system to this forcing. Because of the ~1000-year offset between different records across this period (e.g. Figure 2), we felt it more prudent to minimise the impact of greenhouse gas forcing. We have made a note to this effect in the Methods.

- It is difficult to read the following sentence:

'Following a flux of 0.54 Sv freshwater, all simulations show surface atmospheric cooling over large parts of the Antarctic continent, with significant surface warming (>1.5 °C annual mean temperature increase) and increases (decreases) in sea ice extent in the South Atlantic (Ross Sea

and North Atlantic) accompanied by pervasive warming in the equatorial Pacific (Figure 3).'

Do you really mean a warming in the South Atlantic?

We apologise. This was badly muddled. The text has been revised correctly to the following:

Following a flux of 0.54 Sv freshwater, all simulations show surface atmospheric cooling over large parts of the Antarctic continent, with significant surface warming ($>1.5^{\circ}\text{C}$ annual mean temperature increase) and decreases (increases) in sea ice extent in the Ross Sea and North Atlantic (South Atlantic) accompanied by pervasive warming in the equatorial Pacific (Figure 3).

We thank the reviewer for highlighting this error.

- Abstract: 'Using a fully coupled climate system model we undertake an ensemble of transient meltwater simulations and find that southern salinity anomalies can trigger low-latitude temperature changes that are atmospherically propagated globally via a Rossby wave train, with implications for past and future change.'

Please specify the implications.

The implication is this mechanism may have operated during other periods or in the future. However, this is beyond the scope of the current manuscript so we have deleted this final part of the Abstract.

General Recommendation:

The study is still an interesting contribution for a broader research community and I would be happy to have a look on a revised manuscript version.

Reviewer #3 (Remarks to the Author):

This is my third review of the paper by Turney et al. I have gone through the entire paper once again and read through their response to the reviewers. I have only minor comments for the paper, which the authors can decide to incorporate if they like, as they are not major. Overall, I think the authors have done an outstanding job both addressing and carefully considering all of the reviewer comments, and notably done so very cordially given the reviews (including my own), which asked quite a lot of them and required they do additional analyses and simulations. I personally appreciated this exchange.

We thank the reviewer for their generous comments and helpful advice.

Minor Comments:

Citations 1-5: Wouldn't the most appropriate reference for the bipolar seesaw be Stocker and Jansen, 2003; Broecker, 1998; and/or Stocker, 1998?

These other papers build on the idea but the credit for the idea comes much earlier – in fact the first observations and discussion of the significance (without naming it) comes from the early work of Mix, Ruddiman, and McInyre 1986.

The reviewer is absolutely correct. The suggested articles are more appropriate. We have removed the Broecker (1997) paper and referenced Broecker (1998) and Stocker and Johnsen (2003). We're embarrassed to admit we were not aware of the paper by Mix, Ruddiman and McIntyre (1986). It is a terrible shame that this paper is not as well known as it deserves. It is a remarkably prescient piece of work and we have cited the paper in our manuscript.

Lines 149-150: I'm confused about exactly what exact period the authors are referring. If they are referring to around H3, then I'd say: are you sure there isn't evidence of weakened overturning? The paper by Böhm et al. 2015 Nature (Figure 3) and the Henry et al. 2016 (Figure 2) papers both show a nice summary of overturning proxies and H3 does appear to be demonstrating some reduction.

This was somewhat confusing. We are actually referring to the radiocarbon plateau at ~ 26.4 ^{14}C kyr BP (i.e. SA2). The revised sentence now reads:

'Although there is no evidence for a weakening of circulation across period of the radiocarbon plateau at ~ 26.4 ^{14}C kyr BP (ref. 7-10) (Figure 1), a substantial increase in Antarctic ice-sheet discharge occurred immediately prior to the onset of AIM 4 and is expressed as IRD layer SA2 south of the Polar Front in South Atlantic marine sequences (e.g. TTN057-13)21 (Figures 2 and S6; see Methods).'

REVIEWERS' COMMENTS:

Reviewer #2 (Remarks to the Author):

The authors have done a convincing job to address the remaining comments. Therefore, the interesting paper is recommended for publication.

Response to Reviewer 2 comments

The authors have done a convincing job to address the remaining comments. Therefore, the interesting paper is recommended for publication.

We thank the reviewer for their patience and insightful, balanced comments.